# Designing an App to Promote Physical Exercise in Sedentary People Using a Day-to-Day Algorithm to Ensure a Healthy Self-Programmed Exercise Training

**DOI:** 10.3390/ijerph20021528

**Published:** 2023-01-14

**Authors:** Antonio Casanova-Lizón, José M. Sarabia, Diego Pastor, Alejandro Javaloyes, Iván Peña-González, Manuel Moya-Ramón

**Affiliations:** 1Sports Research Centre, Department of Sport Sciences, Miguel Hernández University of Elche, 03202 Elche, Spain; 2Department of Sport Sciences, Miguel Hernandez University, Alicante Institute for Health and Biomedical Research (ISABIAL), 03010 Alicante, Spain

**Keywords:** app, heart rate variability, short and ultra-short measurements, exercise training

## Abstract

Heart rate variability (HRV) has allowed the implementation of a methodology for daily decision making called day-to-day training, which allows data to be recorded by anyone with a smartphone. The purpose of the present work was to evaluate the validity and reliability of HRV measurements with a new mobile app (Selftraining UMH) in two resting conditions. Twenty healthy people (10 male and 10 female) were measured at rest in supine and seated positions with an electrocardiogram and an application for smartphones at the same time (Selftraining UMH) using recordings obtained through an already validated chest-worn heart rate monitor (Polar H10). The Selftraining UMH app showed no significant differences compared to an electrocardiogram, neither in supine nor in sitting position (*p* > 0.05) and they presented almost perfect correlation levels (r ≥ 0.99). Furthermore, no significant differences were found between ultra-short (1-min) and short (5-min) length measurements. The intraclass correlation coefficient showed excellent reliability (>0.90) and the standard error of measurement remained below 5%. The Selftraining UMH smartphone app connected via Bluetooth to the Polar H10 chest strap can be used to register daily HRV recordings in healthy sedentary people.

## 1. Introduction

Physical activity is an indispensable element for the health of the population. Currently, non-communicable diseases, such as cardiovascular and respiratory diseases, cancer, diabetes, and mental disorders, account for 68% of the causes of death worldwide. Moreover, lack of physical activity increases the risk of diseases, such as obesity, hypertension, the development of different forms of cancer, and cardiovascular diseases [1]. On the other hand, physical activity is a factor that increases the population’s longevity and quality of life [2]. Furthermore, its positive effect on these diseases has led to physical activity being considered as an effective tool for preventing and treating these diseases [3]. Moreover, the American College of Sports Medicine (ACSM) has included physical activity as one of the topics of special relevance among physical activity and health professionals in recent years, coining the term “Exercise is Medicine” [4].

Although supervised exercise has advantages and many people choose to attend supervised interventions to improve their fitness and health, unfortunately, the cost and time involved in travelling to the sport centres is beyond the reach of a large part of the population [5]. For this reason, most individuals who start exercising do so autonomously and without any kind of control [6]. As a result, they rarely adhere to exercising adequately and often put their health at risk [6]. It also means that this practice of physical exercise is not tailored to their needs and can cause an inadequate and inefficient response of the organism due to a lack or an excess of physical exercise which can, as a consequence, limit and even distance them from the benefits they are trying to obtain. To compensate for this, home-based exercise interventions have emerged as a suitable, cheaper and more accessible alternative [6]. However, currently two major problems prevent a real and sustained improvement using home-based exercise interventions, with the consequent quality loss: the low practitioner adherence and the lack of an individualised monitoring strategy.

Adherence is understood as the degree of correspondence between a person’s behaviour and the recommendations provided by health professionals [7]. The factors influencing adherence to exercise and physical activity are complex and range from personal to environmental factors [8]. In addition, we do not have a gold standard for measuring adherence to physical exercise either, which adds further uncertainty in identifying predictors [8]. The increasing development of digital technologies, including wearable technologies (WTs), is enabling to both the collection of objective adherence measures and the improvement of educational processes for the learning of physical exercise practitioners [8]. The WTs include a wide range of devices, which can be embedded in smartphones, smartwatches and other portable devices [9]. Today, most WTs are designed to be easy for users to apply in everyday situations, using wireless technology to transmit information to a device such as a smartphone [9]. Aspects, such as the use of WTs and autonomy support, have been shown to be beneficial in improving adherence to physical exercise [10].

The lack of an individualised monitoring strategy is based on the great heterogeneity that exists in the population in response to the same physical exercise stimulus. We can find different responses to the same physical exercise programmes even in people with a similar fitness and health status level [11]. Therefore, it is necessary to individualise and optimise these physical exercise programmes based on this individual response in order to obtain the greatest health benefits [12]. One tool for assessing an individual’s response to physical exercise is heart rate variability (HRV) [13]. HRV is a non-invasive, valid and reliable measure of the balance of the sympathetic and parasympathetic branch of the autonomic nervous system (ANS) [14]. Numerous studies have identified it as a variable which is sensitive to the effect of training, showing the response to exercise and fatigue levels [13,15] and it has been used to guide training on a daily basis in sports, such as running [16,17,18], cross-country skiing [19], cycling [20,21] or in untrained healthy women [22]. The main findings of these previous studies are that greater increases in physical fitness were obtained for day-to-day models compared to traditional ones [16,19,20] or they showed similar improvements, but with a lower dose of physical exercise by day-to-day models [17,18,21], which meant a time optimisation to achieve the desired improvements. The premise of these day-to-day models is that training is modulated according to the organism’s status (based on HRV), performing vigorous physical exercise when the individual is prepared (i.e., normal values in HRV) and, conversely, performing light physical exercise or resting when there is a sign of excessive fatigue or stress (i.e., alteration in HRV values). Currently, there are some apps that have been validated for HRV measurement using photoplethysmography, such as HRV4Training [23] and Welltory [24], and using a HR monitor with the placement of a Bluetooth-connected chest strap like Elite HRV [25], finding acceptable agreement compared to an electrocardiogram (ECG) as the gold standard. However, these HRV apps indicate the value and whether the subject is in normal condition or not, but they do not recommend physical exercise based on the HRV value.

In order to reduce the problems of home-based exercise interventions and to facilitate access to physical exercise for sedentary people or people with a low level of physical fitness, the authors have developed a mobile application (Selftraining UMH), using a guided day-to-day model based on the HRV parameter root-mean-squared differences of successive RR intervals (rMSSD). Therefore, the aim of this study was to evaluate the validity and reliability of HRV measurements with this new mobile application in two resting conditions (supine and seated) and in short time intervals.

## 2. Materials and Methods

### 2.1. Experimental Design

Two different devices that recorded HRV were used for the measurements: a Biopac MP35 3-lead electrocardiogram (Biopac Systems, Goleta, CA, USA) and a new smartphone application (Selftraining UMH (Elche, Spain)). The electrocardiogram was connected to a computer via a USB port and the mobile application was linked using Bluetooth to a previously validated HR sensor chest strap (Polar H10, Polar Electro Oy, Kempele, Finland) [26]. Four consecutive 6 min recordings were captured by alternating the supine position with the seated position in a resting state [24]. During the 6 min period, the first minute was used to stabilize the HRV signal [27] and the last 5 min in each position were taken for the subsequent analysis.

### 2.2. Participants

Twenty healthy people (10 male and 10 female) participated in this study (their characteristics can be found in Table 1). People with pathologies or physically active people (i.e., people who conducted more than 150 min of moderate physical activity per week) were excluded [28]. Before starting the study, all the participants were informed of the purpose of the study and signed an informed consent form. This study was approved by the Ethics Committee of the university (reference number: CID.DPC.01.19). The study was conducted in accordance with the standards of Good Clinical Practice and international ethical principles applicable to medical research in humans (Declaration of Helsinki).

### 2.3. Data Acquisition

The data collection was conducted under constant temperature and humidity conditions. Participants were instructed to relax and maintain a constant, comfortable respiratory rate using a metronome (range between 7 and 11 breaths per minute), and lights were turned off to ensure a quiet environment that could not alter HRV measurements. The Biopac MP35, considered the gold standard for quantifying the time elapsed between consecutive R-waves (R-R interval) of each heartbeat, was chosen to record the ECG signal [29]. The Biopac MP35 was connected to a computer using the Biopac Lesson Student Lab & for PC software (version 3.7.1), with a sample rate of 1000 Hz. In addition, the Selftraining UMH (version 1.5) smartphone application was used. There are different ways to conduct recordings with the Selftraining UHM app: 1. by photoplethysmography, 2. with a Polar heart rate sensor, 3. through the Welltory mobile app [24], and 4. you can enter the values by hand. In this case, the authors wanted to validate the use of a Polar HR sensor.

### 2.4. Procedures

Prior to recording the HR and HRV, electrode placement on the participant’s chest was distributed according to the recommendations of the American Heart Association (AHA) in the standard ECG position using 3-leads [30]: RA, right midclavicular line, intersection with the second right intercostal space; LA, left midclavicular line, intersection with the second left intercostal space; LL, left midclavicular line, intersection with the last left intercostal space. The participant’s skin was cleaned with alcohol and shaved for the placement of the electrodes. After these steps, the Polar H10 chest strap was placed following the manufacturer’s recommendations. Polar HR monitors have been shown to be suitable for measuring the time between consecutive heartbeats and are not only used in sports, but also in other areas, such as science and medicine for HRV recording and analysis [31]. The chest strap was linked using Bluetooth to the Selftraining UMH application. All the procedures were performed by the same researcher and the measurements were supervised simultaneously on both the devices used. The measurement protocol was repeated in those instances in which synchronization failed. All the recordings were conducted individually for each participant.

After the recordings, the raw R-R interval data files from the two devices were downloaded and examined with Kubios HRV Premium software (version 3.5.0; Biosignal Analysis and Medical Imaging Group, Department of Physics, University of Kuopio, Kuopio, Finland) [32]. For the HRV analysis, the rMSSD parameter was chosen for each measurement period based on its better suitability and reliability than other indices [33]. rMSSD values were transformed to their natural logarithms (LnrMSSD) to allow parametric statistical comparisons assuming normal distributions [34]. Data examination and treatment was performed according to standard criteria [35]. Each file was corrected for ectopic beats and artifacts using an artifact correction method provided by Kubios software (automatic beat correction) prior to the analysis [32]. A mean level of artifact correction that identifies R-R intervals that vary above or below 0.25 s compared to the mean was chosen to help preserve variability and address the presence of any artifacts [25]. The correction of the artefacts was conducted using interpolation methods that allow the replacement of abnormal intervals by a new one. Specifically, Kubios uses cubic spline interpolation which makes the construction of trend curves calculated through a polynomial fit possible [36]. It has been recommended that this artifact correction technique be used for occasional artifacts and ectopic beats when examining R-R intervals [37]. In this study, only recordings with less than 20% of the corrected beats were included, as current literature recommends to maintain at least 80% of the normal consecutive beats for later examination [38]. Consequently, HRV recordings should be examined with caution when lifestyle factors and participant behaviours are not controlled [39].

The data obtained during the five minute measurements in the two devices (ECG and Selftraining UMH) were evaluated in two ways, taking the first minute and the total five minutes, with the aim of comparing the ultrashort duration measurement with the full duration measurement [24].

### 2.5. How the Selftraining UMH Application Works

The application requires a control period of one month, in which participants must measure their HRV (i.e., the parameter LnrMSSD) to establish a profile for each person which constitutes a range of normative values, setting a zone of smallest worthwhile change (SWC), which is used to interpret the changes in values by using a seven day rolling average (LnrMSSD_7day-roll-avg_) for the purpose of prescribing a training based on these values. The SWC is calculated as the mean ± 0.5 × standard deviation (SD) following recommendations from previous studies [40]. The application is integrated to perform three training sessions per week and to perform the day-to-day training prescription based on HRV. It uses the decision making algorithm used by Javaloyes et al. (2019) [20] which is a modification of Kiviniemi et al. (2007) [16]. On the first training day, a low-intensity session is performed. From the first session onwards, when the LnrMSSD_7day-roll-avg_ values remain within the SWC (+), high-intensity training sessions are prescribed. If the LnrMSSD_7day-roll-avg_ falls outside the SWC (−), low intensity or rest sessions are prescribed. Within the application there are four intensity levels and within each intensity level there are four different sessions to choose from. The application is set up as a game, in which by completing sessions of a level the participant obtains stars and when a certain number of stars is reached, the participant progresses to the next level. The application is programmed to perform a training for three months for people who want to start exercising. The physical exercise programme from the Selftraining UMH app is detailed in the Electronic Appendix A (ESM).

### 2.6. Statistical Analysis

To confirm the normality of the data, a Shapiro–Wilk statistical test for a sample of less than 50 subjects was used [41]. A repeated measures ANOVA analysis was performed using the Selftraining UMH app and the Kubios software to evaluate possible differences in means according to the device used or the duration of the measurement (1 min vs. 5 min). In addition, a Bonferroni post hoc test was used to assess the pairwise comparison between the different ways to obtain the measurements. They were expressed in Cohen’s d units and were interpreted as trivial (<0.19), small (0.20–0.49), moderate (0.50–0.79) and large (>0.80), and the effect sizes (ES) were shown with a CI of 95% between groups [42]. Pearson correlation coefficient (r) was calculated to assess the strength of association between HRV measurements with the different devices and the duration of the measurement in 1 min and 5 min and it was interpreted as trivial (<0.09), small (0.10–0.29), moderate (0.30–0.49), high (0.50–0.69), very high (0.70–0.89), and almost perfect (>0.90) [43]. The intraclass correlation coefficient (ICC) [44] and the standard error of measurement (SEM) [45] were used to assess relative and absolute reliability between trials. ICC values were interpreted as poor to moderate (<0.75), good (0.75–0.90), and excellent (>0.90) [46]. The results obtained from the SEM values below 10% were estimated as admissible, and the minimum detectable change (MDC) was evaluated based on the results obtained from the SEM values using the formula 1.96 × SEM × √2 [24]. Bland–Altman analyses were performed on the time domain HRV for LnrMSSD to constitute an agreement between the measurement instruments (Selftraining UMH application vs. ECG), geometrically showing differences and limits of agreement [43]. All the results obtained were analysed using Excel software, a spreadsheet program developed by Microsoft (Microsoft, Seattle, USA) and the JASP program [47]. The results are presented as mean ± SD and the level of statistical significance was set at *p* < 0.05.

## 3. Results

No significant differences (*p* > 0.05) in the data obtained from the Selftraining UMH application in both supine and sitting positions when compared with the ECG (regarded as the gold standard) have been found (Table 2). Selftraining UMH data analysis in the supine position with kubios showed trivial ES (Cohen’s d) while for the smartphone application the ES was moderate. Selftraining UMH data analysis in the sitting position with kubios and with the smartphone showed small ES. Correlation analysis showed a nearly perfect correlation between the Selftraining UMH application and the gold standard (ECG). The scattered plots are shown in Figure 1 and Figure 2 for supine and sitting position, respectively. In the Bland–Altman plots, there was no apparent bias for agreement in the measurement device (Selftraining UMH app) versus the gold standard ECG method in either resting measurement condition (Figure 3).

The repeated measures ANOVA did not show significant differences for the measurement device when comparing the short 5 min measurement and the ultrashort recording of 1 min (Table 3). Furthermore, ES presented trivial effects and the correlation coefficient showed an almost perfect correlation. The scattered plots are shown in Figure 4 and Figure 5 for supine and sitting position, respectively.

Consecutive HRV measurements showed excellent ICC values for reliability analysis (Table 4). Table 4 shows the values for the SEM, reporting values below the 10% limit in all measurements.

## 4. Discussion

The aim of this study was to assess the validity and reliability of HRV measurements with this new Selftraining UMH mobile app in two resting conditions (supine and sitting). To assess validity and reliability, the Selftraining UMH app measurements were compared with the ECG (gold standard).

Selftraining UMH showed no statistical differences (*p* > 0.05) neither in the raw data using the Kubios software, nor in the results analysed by the app in any of the body positions evaluated, compared to the ECG (Table 2). The Selftraining UMH application ES showed trivial to moderate effects for the different body positions in resting conditions; however, as can be seen in Table 2, the statistical analyses showed no differences with respect to the ECG. Furthermore, our study showed an almost perfect correlation (>0.90) in both body positions in resting conditions compared to the ECG in the results analysed with Kubios and with the app (Figure 1 and Figure 2). The results found in the present study are very similar to previous studies with other smartphone applications using a HR monitor with the placement of a Bluetooth-connected chest strap [23,24,25,48]. However, in other studies the correlation coefficient was lower than those shown in this study [24,25,48] and one study showed the same coefficient as in the present study using a HR monitor [23]. Importantly, our study includes a 1 min stabilisation period to obtain valid and reliable HRV measurement data [49] in the time domain, situation which has only been used in one previous study [24].

To analyse validity and reliability, this study analysed the results obtained from ultra-short (1 min) recordings with short (5 min) recordings which are considered the standard. The comparison between ultra-short (1 min) and short (5 min) figures showed no statistical difference in any of the body positions under resting conditions (Table 3). The ES was trivial between both measurements (Cohen’s d < 0.2). For all measurements, the correlation values were almost perfect with a value of r > 0.9 (Figure 4 and Figure 5). The results found in the present study are very similar to previous studies using ECG and other apps that use a chest strap HR monitor to measure HRV [23,24,25,48]. Furthermore, the Bland–Altman plots revealed that the Selftraining UMH app shows good agreement in both body positions under resting conditions for the magnitudes of bias and 95% CI (Figure 3) compared to the ECG. The limits of agreement and confidence intervals obtained for the bias show that there is no statistical difference for the application of the Selftraining UMH vs. ECG system between measurement of the two body positions under resting conditions.

In this study, an excellent intraclass correlation coefficient (>0.90) together with a standard error of measurement below 10% suggested the reliability of the ECG in both body positions. In resting conditions, using the ECG is the method normally used to measure HRV. It should be noted that the values obtained in the supine position (0.93) are lower than the values obtained in the sitting position (0.99). Since both supine and seated values are considered to have good reliability, both positions can be recommended to assess HRV in healthy sedentary people, with an emphasis on always using the same position. Previous research in highly endurance-trained athletes and populations with low resting HR has recommended the seated position, as with this position it is possible to avoid the phenomenon called parasympathetic saturation which makes HRV trends more difficult to interpret and affects measurements [50].

Previous studies have favoured the use of a chest strap to measure HRV as there is no possibility of momentary disconnection between the skin and the electrodes [26]. In addition, for those who are not familiarised with the use of current technology such as photoplethysmography, there may be differences attributed to finger size, position, skin characteristics and pressure exerted on the sensor [51,52,53].

The validation of the Selftraining UHM application will allow the collection of HRV data within the time domain (rMSSD) in a sample of different age ranges to generalise and compare them with other existing applications, being able to create reference values for healthy sedentary people. Therefore, it will be possible to complete, update and extend previously published databases and to define reference values for age groups under 20 years of age and over 60 years of age according to sex, which these databases do not report or which have a low sample number [54,55,56]. Furthermore, it cannot be ruled out that in the future the data obtained from the Selftraining UMH application will be consistent with healthy people in other regions of the world. It should also be noted that ECGs in many previous studies were performed at a sampling rate of 250 Hz or lower, but now devices with higher sampling rates (≥1000 Hz) are available. Although several studies have shown that the use of lower sampling rates could be used without significantly affecting HRV indices [54,57,58], the accuracy of measurements at higher sampling rates (1000 Hz) is perfect [59]. Furthermore, future research of the project thanks to the validation of the Selftraining UMH application will aim to compare autonomously HRV-guided training through this mobile application with HRV-guided training through physical activity professionals and traditional training.

This study demonstrated that there was no apparent matching bias in the measurement device (Selftraining UMH app) in the Bland–Altman plot results and the observed relationship expressed through Pearson product-moment correlation was almost perfect against the gold standard ECG method in both measurement conditions (supine and seated). These results may encourage practitioners to implement these applications for long-term HRV-based monitoring. The results shown in this study follow that the Selftraining UMH app is valid for short and ultra-short HRV recordings using the heart rate monitor with placement of a Bluetooth-connected chest strap. To the authors’ knowledge, this is the only validated app that uses variability as a control using a guided day-to-day model based on the HRV parameter (rMSSD) as current apps that measure HRV only indicate the rMSSD value and whether or not the tested subject is in normal conditions.

Following recommendations from previous studies, a correction for artefacts and ectopic beats was performed [25,27]. However, the Selftraining UMH application uses a Software Development Kit (SDK) provided by Polar which they offer for free and with an easy access to its API (Application Programming Interface). Therefore, it is possible that the HRV signal analysis contributed to a bias by not being able to compare the signal correction methods used by the different applications [24]. Future research can explore the impact of the different artifacts and ectopic beat correction methods used by previously validated smartphone apps to corroborate adequate signal analysis processing when measuring HRV. Second, lifestyle factors (diet, sleep, smoking, alcohol intake, etc.) are known to affect HRV [55] and were not controlled for. However, the focus of the study was not to investigate the interaction between these variables and HRV.

## 5. Conclusions

To the authors’ knowledge, with respect to the smartphone app, this is the first study to assess the validity and reliability of Selftraining UMH for measuring HRV within the time domain (rMSSD). The results shown in this study presented similar levels of reliability to the ECG (gold standard) with excellent values of intraclass correlation coefficients and all standard error values of the measurement were below 10%, the highest value being 4.66 (Table 4). Therefore, it appears that the Selftraining UMH smartphone app connected via Bluetooth to the Polar H10 chest strap can be used to register daily HRV recordings in healthy sedentary people.

## Figures and Tables

**Figure 1 ijerph-20-01528-f001:**
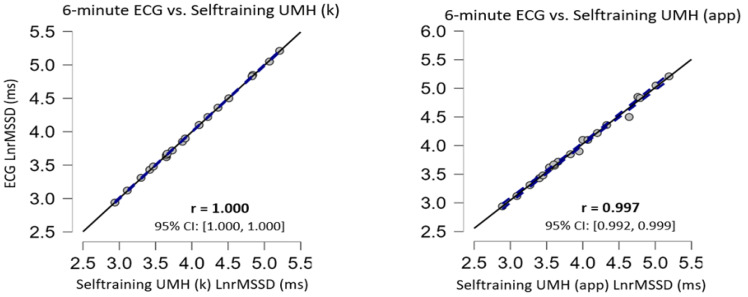
Correlation plots in the supine position, the confidence intervals of ±95% are expressed by dashed lines and the solid black line represents the line of equivalence (r = 1.0).

**Figure 2 ijerph-20-01528-f002:**
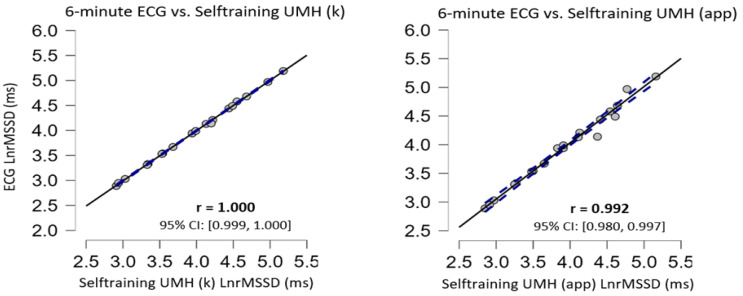
Correlation plots in the sitting position, the confidence intervals of ±95% are expressed by dashed lines and the solid black line represents the line of equivalence (r = 1.0).

**Figure 3 ijerph-20-01528-f003:**
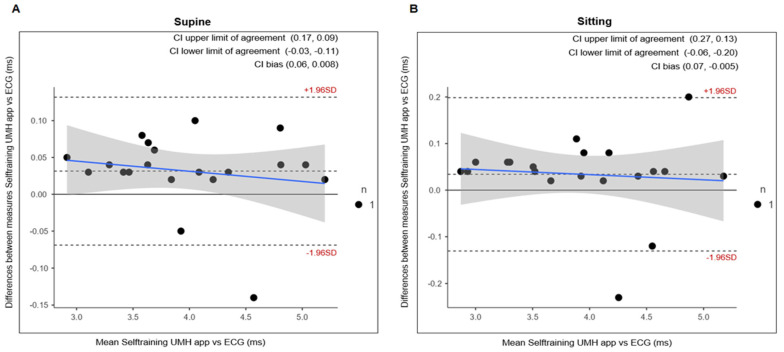
Comparing Bland–Altman plots in rMSSD. The images included show the values between ECG and Selftraining UMH in supine (**A**) and sitting (**B**) position.

**Figure 4 ijerph-20-01528-f004:**
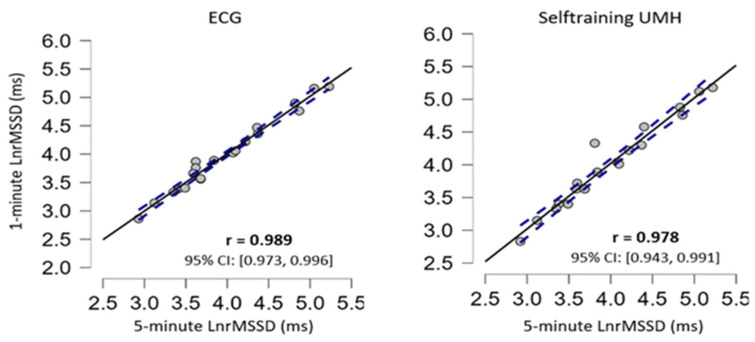
Correlation plots in the supine position, the confidence intervals of ±95% are expressed by dashed lines and the solid black line represents the line of equivalence (r = 1.0).

**Figure 5 ijerph-20-01528-f005:**
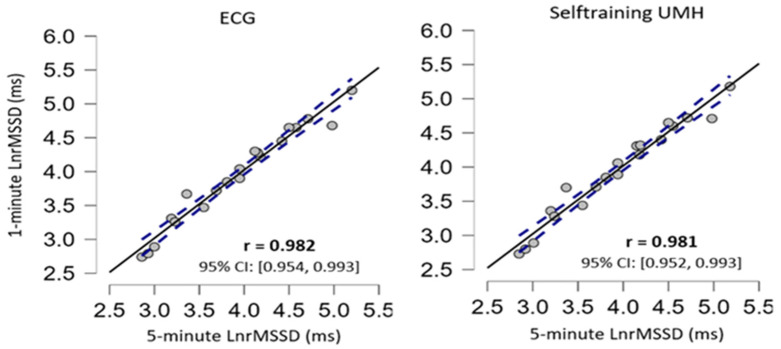
Correlation plots in the sitting position, the confidence intervals of ±95% are expressed by dashed lines and the solid black line represents the line of equivalence (r = 1.0).

**Table 1 ijerph-20-01528-t001:** Participants descriptive statistics (mean ± SD).

	Males (*n* = 10)	Females (*n* = 10)
Age (y)	27.70 ± 4.40	25.00 ± 2.67
Weight (kg)	77.10 ± 4.71	62.43 ± 5.91
Height (m)	1.77 ± 0.05	1.64 ± 0.08
BMI (kg∙m^−2^)	24.54 ± 1.76	23.27 ± 2.08
Breathing frequency (bpm)	9.50 ± 0.94	9.70 ± 1.06

N, sample number; SD, standard deviation; Y, years; M, meters; Kg, kilograms; Bpm, breaths per minute.

**Table 2 ijerph-20-01528-t002:** Comparison of the ECG vs. the Selftraining UMH app.

Position	Device	Descriptive Data	MD	*p*	Effect Sizes (95%CI)
Supine	ECG	3.995 ± 0.64			
	Selftraining UMH (kubios)	3.998 ± 0.65	−0.002	1.00	−0.174 (−0.010, 0.006)
	Selftraining UMH (app)	3.964 ± 0.65	0.031	0.13	0.615 (−0.005, 0.068)
Sitting	ECG	3.947 ± 0.66			
	Selftraining UMH (kubios)	3.951 ± 0.66	−0.004	1.00	−0.236 (−0.016, 0.008)
	Selftraining UMH (app)	3.913 ± 0.67	0.034	0.86	0.405 (−0.026, 0.094)

ECG, electrocardiogram; app, smartphone application analysis; MD, Mean Difference; CI, Confidence intervals.

**Table 3 ijerph-20-01528-t003:** Comparison between ultra-short 1 min and 5 min standard recordings.

Device	Position	1 min	5 min	MD	*p*	Effect Sizes (95%CI)
ECG	Supine	3.978 ± 0.66	3.970 ± 0.65	0.007	0.73	0.078 (−0.038, 0.053)
	Seated	3.942 ± 0.71	3.918 ± 0.69	0.025	0.42	0.183 (−0.038, 0.087)
Selftraining UMH	Supine	4.002 ± 0.66	3.981 ± 0.65	0.020	0.52	0.147 (−0.045, 0.086)
	Seated	3.939 ± 0.70	3.920 ± 0.69	0.019	0.52	0.145 (−0.043, 0.082)

MD, Mean Difference; CI, Confidence intervals.

**Table 4 ijerph-20-01528-t004:** Reliability of the ECG and Selftraining UMH.

Position	Device	Mean Difference	ICC (90%CI)	SEM (%)	MCD (%)
Supine	ECG	0.10 ± 0.26	0.93 (0.82, 0.97)	4.51	0.50
	Selftraining UMH (kubios)	0.09 ± 0.26	0.92 (0.82, 0.97)	4.54	0.50
	Selftraining UMH (app)	0.09 ± 0.26	0.92 (0.82, 0.97)	4.66	0.51
Sitting	ECG	0.03 ± 0.12	0.99 (0.97, 0.99)	2.11	0.23
	Selftraining UMH (kubios)	0.03 ± 0.11	0.99 (0.97, 1.00)	1.99	0.22
	Selftraining UMH (app)	0.01 ± 0.11	0.99 (0.97, 1.00)	1.98	0.21

ICC, Intraclass correlation coefficient; SEM, Standard error of the measurement, MDC: Minimal detectable change; app, smartphone application analysis.

## Data Availability

The datasets generated from the current study are available from the corresponding author on reasonable request.

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
