# Peer review of "Designing an App to Promote Physical Exercise in Sedentary People Using a Day-to-Day Algorithm to Ensure a Healthy Self-Programmed Exercise Training"

_ijerph, 2023, doi:10.3390/ijerph20021528_

Round 1

Reviewer 1 Report

The research idea is interesting, but with such a small group size, it should be treated as a pilot study.

 I did not know what activities the respondents participated in? Was the activity undertaken individually or collectively? It should be developed.

In table 1, information characterizing women and men should be provided separately. Average body weight, body height or BMI cannot be given together without taking into account the gender factor. Gender distinctions also need to be included in the charts.

There is no description of what the programmed physical training consisted of and how it proceeded.

 On what basis was it concluded that HRV records apply to healthy people with a low level of fitness?

Reviewer 2 Report

Overall quality of current paper is average.

Among MAIN POINTs of WEAKNESS we can list:

English language must be checked. Sentences such as at lines 13-14 should be totally re-written.

Moreover, Table 1 might be ameliorated (graphically check for typing mean and...).

Finally, I'd like to suggest the following articles in order to improved global discussion and probably insert in references

J Med Internet Res. 2019 Nov 28;21(11):e14343. doi: 10.2196/14343.

JMIR Mhealth Uhealth. 2017 Aug 9;5(8):e119. doi: 10.2196/mhealth.6974.

BMJ Open. 2020 Dec 15;10(12):e040479. doi: 10.1136/bmjopen-2020-040479.

About publication priority I think it is not high especially because of sample number is quite short. Therefore, Editors should evaluate this item.

Best regards.

Round 2

Reviewer 1 Report

My question ("Was the activity undertaken individually or collectively?") was about the type of physical activity undertaken, not how it was measured. The authors did not answer my question.

If the authors cannot provide a description of the programmed training, the use of this wording in the title of the article is unjustified and should be removed from the title.

The authors did not convince me that the gender factor can be omitted in this type of measurement. This falsifies the measurement. I believe that the results should be given in two different groups - separately for women and separately for men. What's the point of listing the combined BMI, weight and height for both sexes together. The third column in Table 1 is therefore redundant.

I have not received an answer to the question "On what basis have HRV records been determined to be for healthy people with a low level of fitness?".
